# Low-Vacuum Pyrolysis of YBCO Films by Using Fluorine-Free Metal Organic Chemical Deposition

Zhao Yang [1] , Chuanbing Cai [1,2,*], Ningdong Chu [1], Shuyun Tong [1], Yuming Lu [1,2] and Zhiyong Liu [1,2]

1    Shanghai Key Laboratory of High Temperature Superconductors, Department of Physics, Shanghai University, Shanghai 200444, China; yangzhaoshu@shu.edu.cn (Z.Y.); cnd152018@shu.edu.cn (N.C.); 3134902515@shu.edu.cn (S.T.); ymlu@shu.edu.cn (Y.L.); zyliu@shu.edu.cn (Z.L.)
2    Shanghai Frontiers Science Center of Quantum and Superconducting Matter States, Department of Physics, Shanghai University, Shanghai 200444, China
*    Correspondence: cbcai@t.shu.edu.cn

**Abstract:** The preparation of YBCO superconducting films by using metal organic chemical deposition (MOD) involves low-temperature pyrolysis and high-temperature treatment. The former process generally requires the introduction of water vapor and other gases. The study on pyrolysis in a low vacuum environment and non-carrier gas atmosphere has never been reported. In this work, we explored a low vacuum pyrolysis scheme with simple Argon gas decompression and a carrier-free atmosphere. The effects of heating rate on the microstructure of pyrolysis films were investigated, and the high-temperature treatment temperature ($T_h$) was also optimized. Compared with conventional pyrolysis, the present low-vacuum pyrolysis does not employ the flowing dry or wet gases, facilitating the internal gas release during film decomposition. More importantly, the efficiency was greatly improved with reduced pyrolysis time. The obtained film surface is free of CuO particle, which leads to a lower roughness. We also investigated the effect of $T_h$ on the final YBCO film texture and superconductivity. As $T_h$ increased from 810 °C to 815 °C, the BaCuO$_2$ phase decreased with enhanced $c$-axis orientation being evident by XRD and Raman spectra. As a result, the critical current density ($J_c$) increased from 0.38 MA/cm$^2$ to 1.2 MA/cm$^2$ (77 K, self-field).

**Keywords:** FF-MOD; low vacuum; YBCO film; epitaxial growth; critical current density



## 1. Introduction

High-temperature superconducting REBa$_2$Cu$_3$O$_y$ (REBCO, RE = Y, Gd, etc.) coated conductors play a crucial role in power transmission and nuclear fusion reactors due to their high operating temperature and current-carrying capacity in the field [1,2]. The existing preparation methods mainly include pulsed laser deposition (PLD) [3], chemical vapor deposition (CVD) [4], reactive co-evaporation deposition (RCE) [5], and metal organic chemical deposition (MOD) [6–8]. Fluorine-free metal organic chemical deposition (FF-MOD) became the current trend owing to its environmental friendliness, low overall cost, and high growth rate [9]. The preparation process consists of low-temperature pyrolysis and high-temperature treatment [8]. The reported pyrolysis processes were carried out at normal pressure with a continuous flow of dry or wet gases, such as Ar, O$_2$, the mixture of N$_2$ and O$_2$ [2]. This process is related to the pyrolysis reaction between acetate and organic thickeners. The University of Tokyo reported the pyrolysis of precursor films at atmospheric pressure (500 °C, 120 min). Later, Southwest Jiao tong University reported the conventional pyrolysis at a lower ramp rate of 0.5 °C/min at atmospheric pressure [10]. Recently, the Barcelona team proposed liquid-assisted ultrafast growth of superconducting films derived from chemical solutions (TLAG), which significantly improved the growth rate of superconducting layers through fast liquid-phase assisted growth [9]. Unfortunately, vacuum fast is only present in high-temperature growth, while the preparation stays

complicated with excessive time. Table 1 summarizes the characteristics of different research institutions in the low-temperature pyrolysis stage [9–17].

**Table 1.** Technical characteristics of different research institutions in low-temperature pyrolysis using by FF-MOD.

| Institution. /Year | Substrate | Flowing Gas | Pressure in Tube Furnace | Heating Rate (°C/min) | Temperature Range (°C) | Total Pyrolysis Time (min) |
|---|---|---|---|---|---|---|
| Southwest Jiaotong University /2015 | $LaAlO_3$ | Humid $O_2$ + Ar mixture | Atmospheric pressure | 0.5 | 110–500 | >720 (con) |
| University of Tokyo /2014 | $SiTro_3$ | Humid $O_2$ | Atmospheric pressure | —— | 500–500 | 120 |
| Tokyo Metropolitan University /2020 | $LaAlO_3$ | Air | Atmospheric pressure | —— | 600–600 | 30 |
| Technical University of Denmark /2015 | $LaAlO_3$ | Humid $O_2$ | Atmospheric pressure | 10 | 20–450 | 43 |
| Shanghai Jiao Tong University /2021 | $CeO_2$ | Humid $O_2$ | Atmospheric pressure | 10 | 100–500 | 40 |
| University of Barcelona /2020 | $LaAlO_3$ | Humid $O_2$ | Atmospheric pressure | 3 | 240–500 | >120 |

This paper first applied low-vacuum technology in the stage of low-temperature pyrolysis. By simply depressurizing the air to 5 Pa, low vacuum pyrolysis films different from the normal pressure pyrolysis films were obtained, avoiding the use of pyrolysis gases and greatly reducing the time. On this basis, the high-temperature treatment ($T_h$) was optimized to achieve good epitaxial growth of YBCO superconducting layers.

## 2. Materials and Methods

The precursor solution was prepared by dissolving $Y(CH_3COO)_3 \cdot 4H_2O$ (99.9%), $Cu(CH_3COO)_2 \cdot H_2O$ (99.9%) and $Ba(CH_3COO)_2$ (99.8%) in propionic acid ($CH_3CH_2COOH$, 99.9%) according to the stoichiometric ratio $n(Y):n(Ba):n(Cu) = 1.3:2:3.6$. After magnetic stirring at 60 °C for 5 h, organic thickener PVB (polyvinylpyrrolidone, 99.7%) was added to the solution. Then, continuously magnetically stirred at 40 °C for 24 h to obtain the final YBCO precursor solution (1.5 mol/L), which was used to prepare superconducting films (100–150 nm thick).

The decomposition process of the metal-organic precursors was determined by differential thermal/thermogravimetric analysis (DTA/TGA) of the dried powders. X-ray diffraction (XRD, Bruker-D2) was applied to characterize the phase composition. The surface microstructure and phase purity of the films were characterized by scanning electron microscopy (SEM, HITACHI-SU5000) equipped with energy dispersive spectroscopy (EDS, HITACHI-SU5000), atomic force microscopy (AFM, Bruker Dimension Edge), and Raman spectroscopy (Raman, RENISHAW-INVIA). The four-point probe method was used in the superconducting transition characteristics ($T_c$, Temperature range: 273 K–4 K). The $J_c$ were determined with the $J_c$ -Scan method ($J_c$ -Scan, THEVA, 77 K, self-field).

## 3. Results

### 3.1. DTA/TGA Analysis of the Precursor Materials

The DTA/TGA of three acetates and PVB is shown in Figure 1. The atmosphere and protective gas of TGA/DTA analysis are argon gas, with flow rates of 10 mL/min and 20 mL/min, respectively. The chemical Equations (1) and (2) represent the loss of crystal

water that occurred in $Y(CH_3COO)_3 \cdot 4H_2O$ (135 °C) and $Cu(CH_3COO)_2 \cdot H_2O$ (170 °C) before 200 °C.

$$Y(CH_3COO)_3 \cdot 4H_2O \rightarrow Y(CH_3COO)_3 + H_2O \uparrow \tag{1}$$

$$Cu(CH_3COO)_2 \cdot H_2O \rightarrow Cu(CH_3COO)_2 + H_2O \uparrow \tag{2}$$

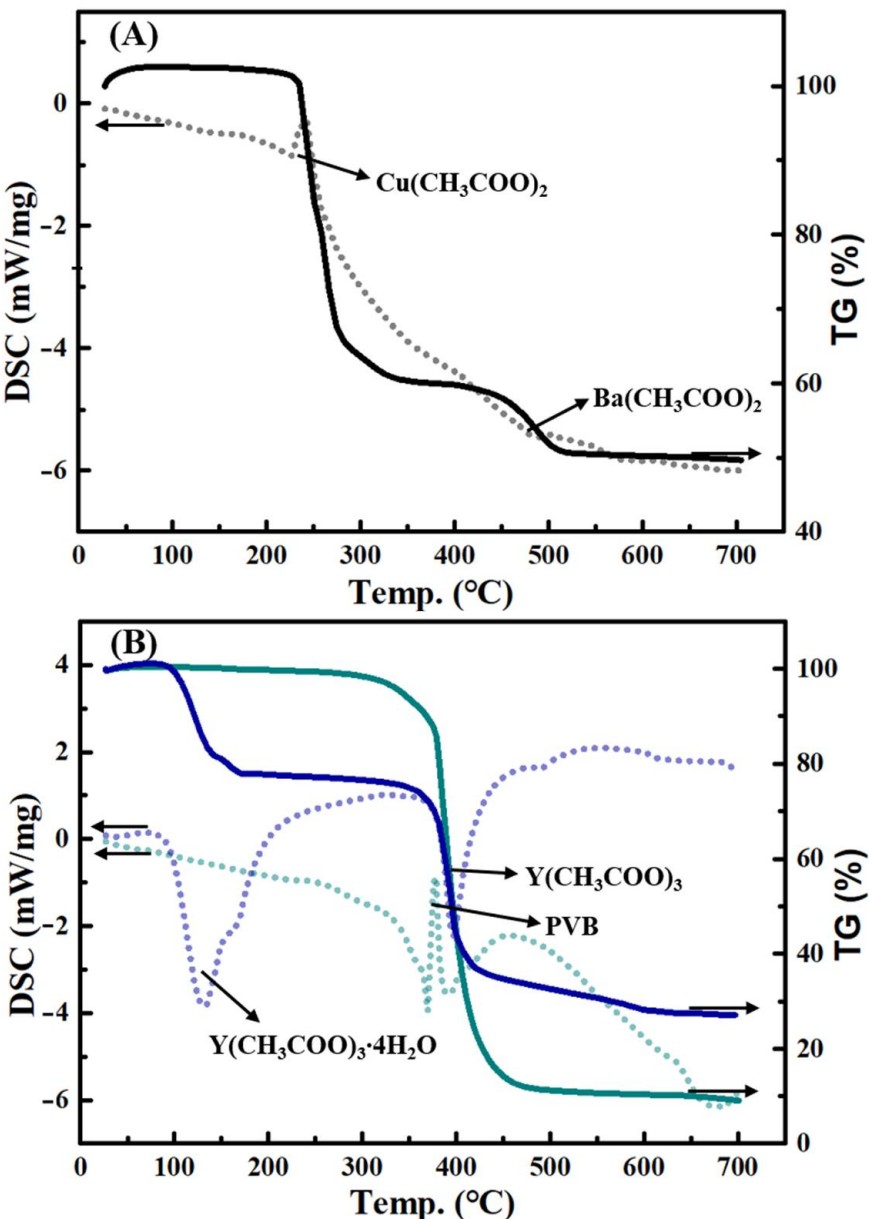

**Figure 1.** DTA/TGA of three acetates and PVB in argon gas at the heating rate of 20 °C/min. (**A**) Mixture of $Ba(CH_3COO)_2$ and $Cu(CH_3COO)_2 \cdot H_2O$; (**B**) $Y(CH_3COO)_3 \cdot 4H_2O$ and PVB. The solid line is DSC, the dotted line is TG.

The decomposition reactions of $Cu(CH_3COO)_2$ (284 °C), $Y(CH_3COO)_3$ (398 °C), $Ba(CH_3COO)_2$ (460 °C) and PVB (375 °C) are shown in Equations (3)–(6), respectively (Figure 1A,B).

$$Cu(CH_3COO)_2 \rightarrow CuO + CO_2 \uparrow + H_2O \uparrow \tag{3}$$

$$Y(CH_3COO)_3 \rightarrow Y_2O_3 + CO_2 \uparrow + H_2O \uparrow \tag{4}$$

$$Ba(CH_3COO)_2 \rightarrow BaCO_3 + CO_2 \uparrow + H_2O \uparrow \tag{5}$$

$$PVB \rightarrow CO_2\uparrow + H_2O \uparrow \qquad (6)$$

The final composition of the precursor film are $CuO$, $Y_2O_3$ and $BaCO_3$ [18–20]. During the high-temperature treatment, BaCuO is generated at 680 °C in a mixture of $N_2$ and $O_2$ [13,21]. It was found that the mixture of $Ba(CH_3COO)_2$ and $Cu(CH_3COO)_2$ thermally decomposed at 700 °C and were nonreactive in their products (Figure 1A). In this study, the pyrolysis temperature was increased to 600 °C, leading to more complete pyrolysis.

Wet films were prepared on LAO single crystal substrates by spin coating (3500 r/min, 20 s). Then, they were placed in an infrared drying oven (110–135 °C, 20 min, 2% humidity) to obtain propionic acid-free sol-gel films, which hindered the generation of cracks.

The low-vacuum pyrolysis was carried out in a tube furnace connected to a mechanical vacuum pump. The pressure in the tube furnace stayed at 5 Pa after argon cleaning (Supplementary Figure S1). No more gas was introduced until the end of the pyrolysis. As illustrated in Figure 2A, pyrolysis was accomplished by heating to 600 °C at a constant rate of 10–25 °C/min. In the stage of high-temperature treatment, a mixture of $N_2$ and $O_2$ (200 ppm) was introduced at 400 mL/min to restore normal pressure. Meanwhile, the temperature was increased to 810 °C or 815 °C at 20 °C/min. The process lasted for 30 min before cooling down naturally. At 600 °C, $O_2$ was introduced (400 mL/min) and kept at 450 °C for 60 min to compensate for the previous oxygen consumption. YBCO transformed from tetragonal phase to orthogonal phase while having superconducting properties. A conventional atmospheric pressure pyrolysis scheme requiring flowing Argon gas is shown in Figure 2B. This paper proposed a low-vacuum pyrolysis scheme and investigated the surface microstructure of the pyrolysis film. With the optimized heat treatment temperature, YBCO superconducting films with good epitaxial growth were successfully prepared.

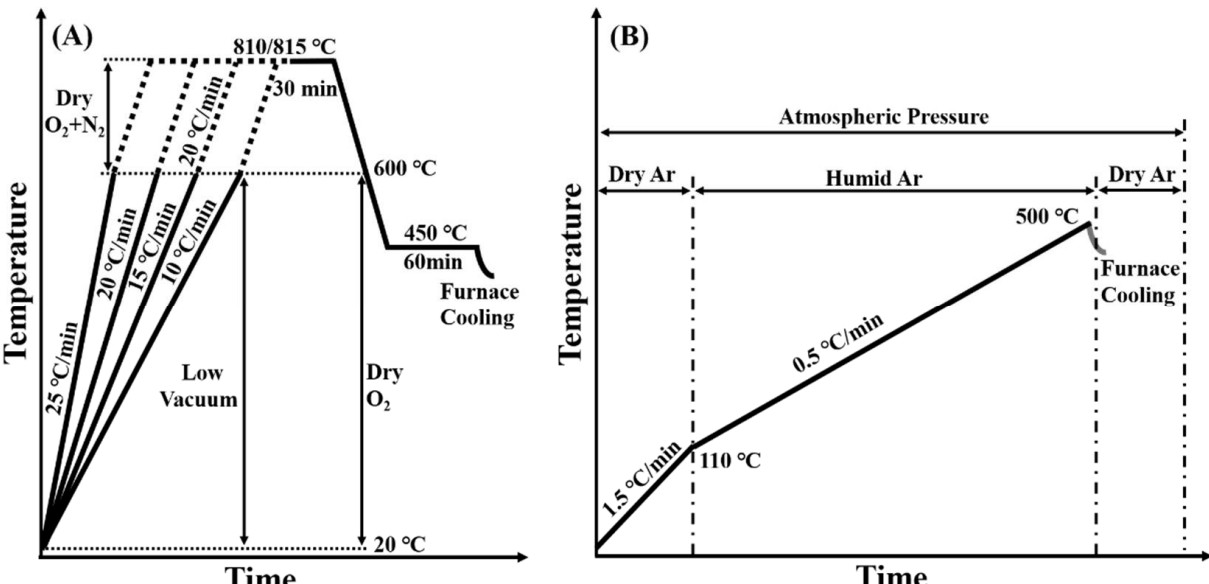

**Figure 2.** (**A**) Heating profile of low-vacuum pyrolysis YBCO film, (**B**) conventional film by FF-MOD.

### 3.2. Characteristics of the Low-Vacuum Precursor Films

The X-ray diffraction patterns of low-vacuum pyrolysis film shown in Figure 3 exhibit similar weak peaks of $BaCO_3$ regardless of the heating rate. In comparison, the conventional film shows stronger peaks of $BaCO_3$ and $CuO$, probably because the grain growth was promoted by a longer pyrolysis time [22]. In LAO substrate, the peaks of $CuO$ and $Y_2O_3$ may be covered by the intense peaks of LAO substrate or $BaCO_3$, $CuO$ and $Y_2O_3$ exist as smaller amorphous particles in a low vacuum environment.

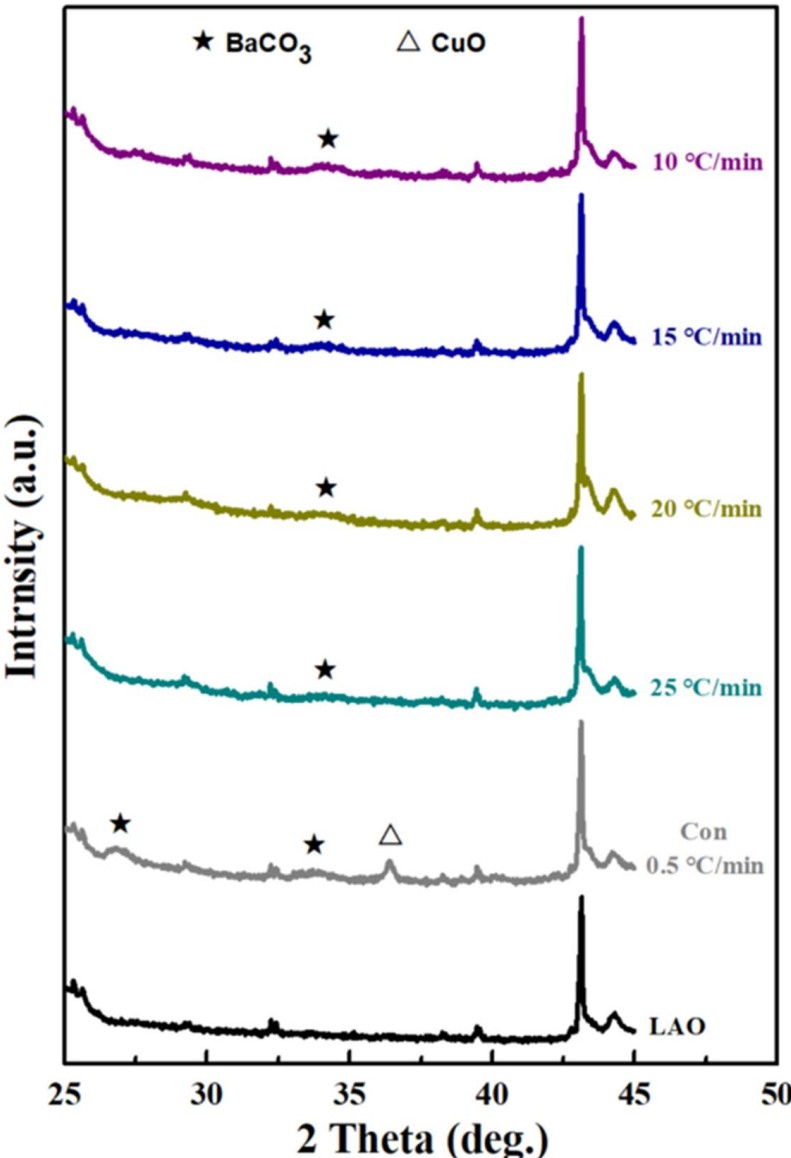

**Figure 3.** The $\theta$–$2\theta$ X-ray diffraction patterns of low-vacuum pyrolysis film, conventional film (Con) and LaAlO$_3$ substrate. The heating rates are indicated on the right side.

Figure 4 shows the representative secondary electron images of low-vacuum films and conventional film. As the heating rate increases from 10 °C/min to 15 °C/min, the vanished folds lead to a smoother surface (Figure 4A, Supplementary Figure S2). Besides, large amount of particles with smaller sizes appear on the surface. Comparing the samples at 15 °C/min and 20 °C/min, the two surfaces show similar morphology. We continued to increase the heating rate to 25 °C/min, and an extremely rough surface was obtained (Figure 4B). In contrast, the surface of the conventional film shows a morphology without cracks and cavities but with large particles (Figure 4C). In the low-vacuum pyrolysis process, the decomposition of organic matter inside the film generates a large amount of H$_2$O and CO$_2$, which are likely to diffuse by pressure difference and temperature rise. Pyrolysis is promoted due to the quickly released stress inside the film [17,18]. The heating rates of 15 °C/min and 20 °C/min may be more favorable for the diffusion of gases, therefore obtaining better surface morphology. Samples for subsequent tests were prepared at 15 °C/min.

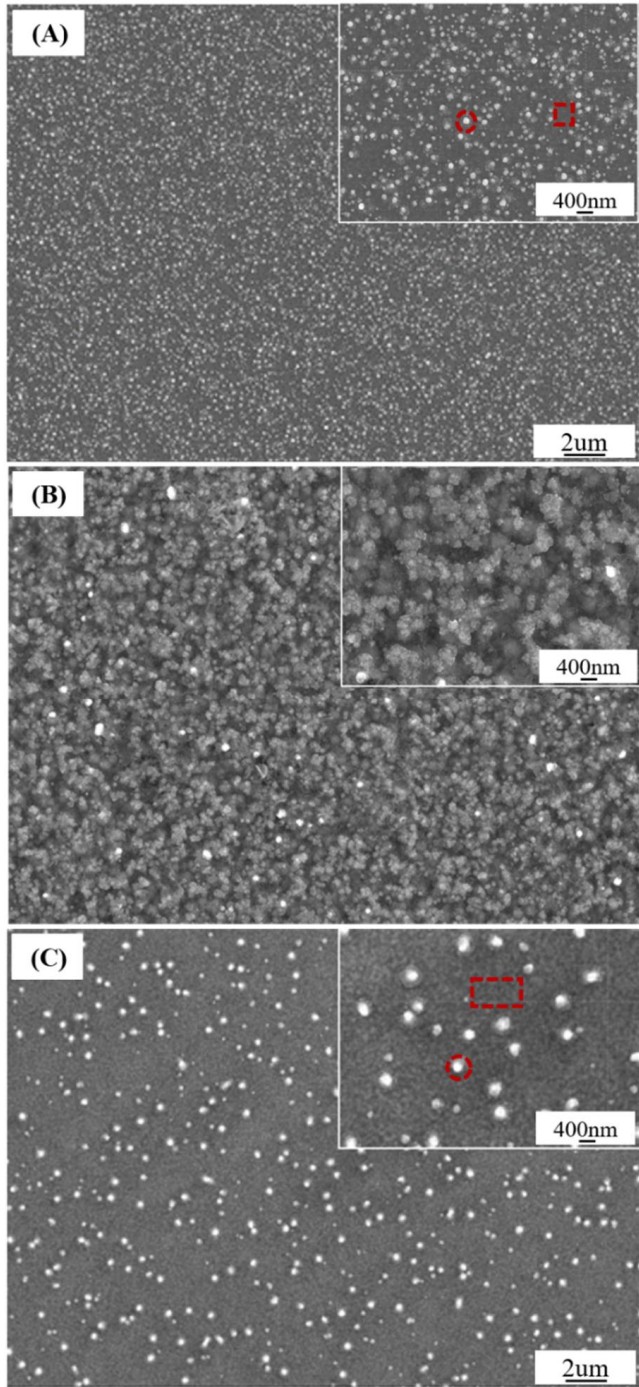

**Figure 4.** Secondary electron images of low vacuum pyrolysis films: (**A**) 15 °C/min; (**B**) 25 °C/min. Secondary electron images of conventional pyrolysis films: (**C**) 0.5 °C/min. The elemental proportions of red circle and red box in Figure (5B,5D).

To investigate the surface morphology of low-vacuum film (15 °C/min) and conventional film, elemental distribution analysis and surface particle size distribution were performed. As illustrated in Figure 5A, about 76.5% of the particles in the low-vacuum film were less than 50 nm, and 10.9% of the particles were 50-100 nm. The EDS analysis suggests similar ratios of elements in granular and non-granular regions (Figure 5B). Compared with low-vacuum film, about 22.9% of the particles in the conventional film were 300–500 nm, and 3.1% of the particles reached 500–600 nm (Figure 5C). The EDS analysis shows obvious difference in elemental proportions between granular and non-granular

regions in conventional film (Figure 5D). According to the studies, the particles in the granular region with 19.3% of Cu and very little Ba, Y should be CuO [13,22]. Notably, the smaller particles on the surface of the low-vacuum film are not CuO. The elemental distribution is more consistent throughout the pyrolysis film, which may facilitate the formation of YBCO superconducting film.

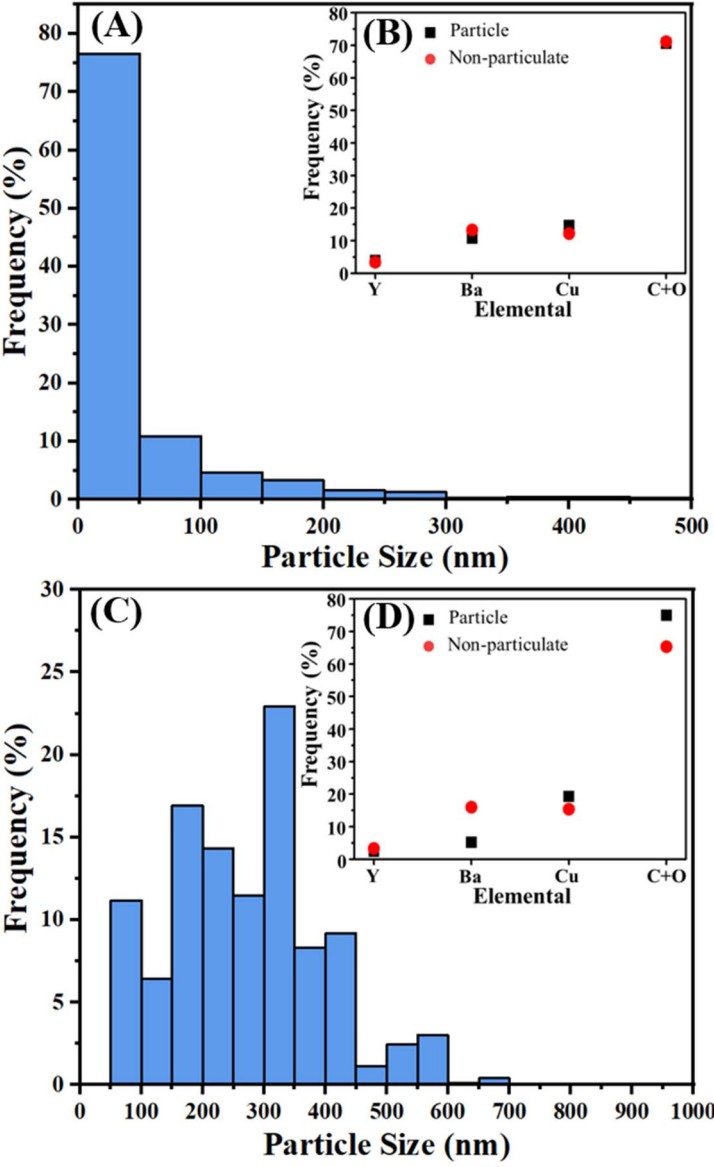

**Figure 5.** Statistical histograms of surface particle size for low vacuum pyrolysis film and conventional pyrolysis film, and elemental proportions for EDS measurements. (**A**) Statistical histograms of surface particle size for low vacuum pyrolysis film (15 °C/min), (**B**) elemental proportions of surface granular and non-granular regions. (**C**) Statistical histograms of surface particle size for conventional pyrolysis film (0.5 °C/min), (**D**) elemental proportions of surface granular and non-granular regions.

During the pyrolysis process, the simultaneous presence of large mass loss and multiple decomposition reactions lead to more than 50% volume shrinkage and release of inhomogeneous strain, thus affecting the film roughness. We compared the AFM images of low-vacuum film and conventional film to investigate the possible reasons for the difference in roughness (Figure 6A,C). The surface roughness Rq of the low-vacuum film is 14.8 nm, much smaller than the 29.3 nm of conventional film. The corresponding height profiles along the diagonal are shown (Figure 6B,D). It was found that the presence of a few

particles increased the roughness of the low-vacuum film. One of the possible explanations for the difference in roughness is that the short pyrolysis time of low-vacuum film limits the growth of film particles. Besides, gases generated by the internal decomposition of the film and inhomogeneous strain are easily released under a low vacuum.

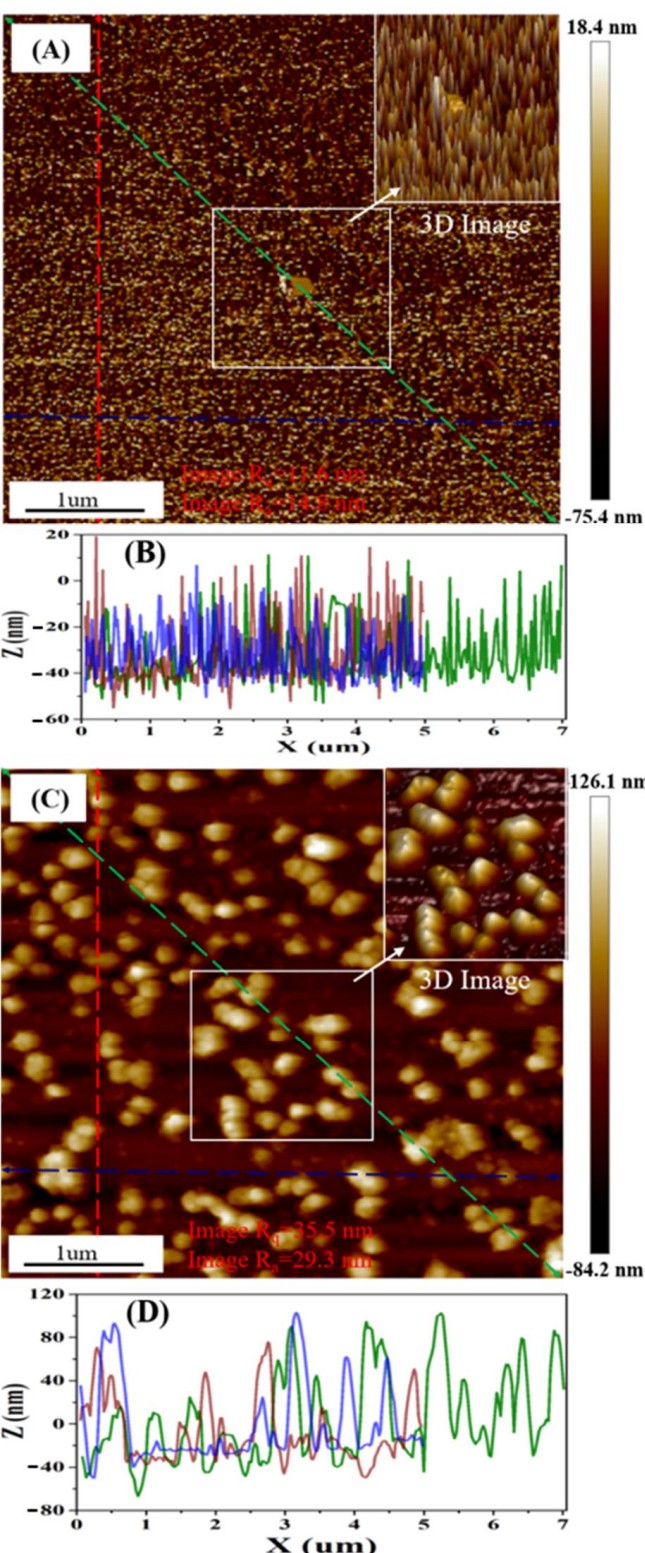

**Figure 6.** (**A**,**C**) AFM images of low vacuum pyrolysis film (15 °C/min) and conventional pyrolysis film (0.5 °C/min). (**B**,**D**) The horizontal, vertical, and diagonal height curves are plotted in blue, red, and green lines.

### 3.3. Optimization of High-Temperature Heat Treatment

The secondary electron images of YBCO films obtained by high-temperature treatment of low-vacuum films and conventional film are shown in Figure 7. The high-temperature treatment temperature $T_h$ of 810 °C leads to a relatively rough film surface, having pores and dendritic grains with apparent boundaries (Figure 7A). The composition of the dendrites was analyzed by EDS spectrum (Supplementary Figure S3A). It shows a relatively high and close proportion of Ba and Cu elements, indicating grains are probably BaCuO$_2$ [9]. These grains are mostly generated in the early stage of YBCO epitaxial growth and incomplete reaction [14]. When $T_h$ was 815 °C, a smoother film with a few cavities and particles on the surface was obtained (Figure 7B). A certain amount of C element appears in the EDS spectrum of the particles but is unseen in the other two samples, which suggests the particles as an impurity phase (Supplementary Figure S3B). In contrast, there are more particles on the surface of conventional YBCO film (Figure 7C). After analyzing the ratio of elements, we considered the particles are BaCuO$_2$ (Supplementary Figure S3C).

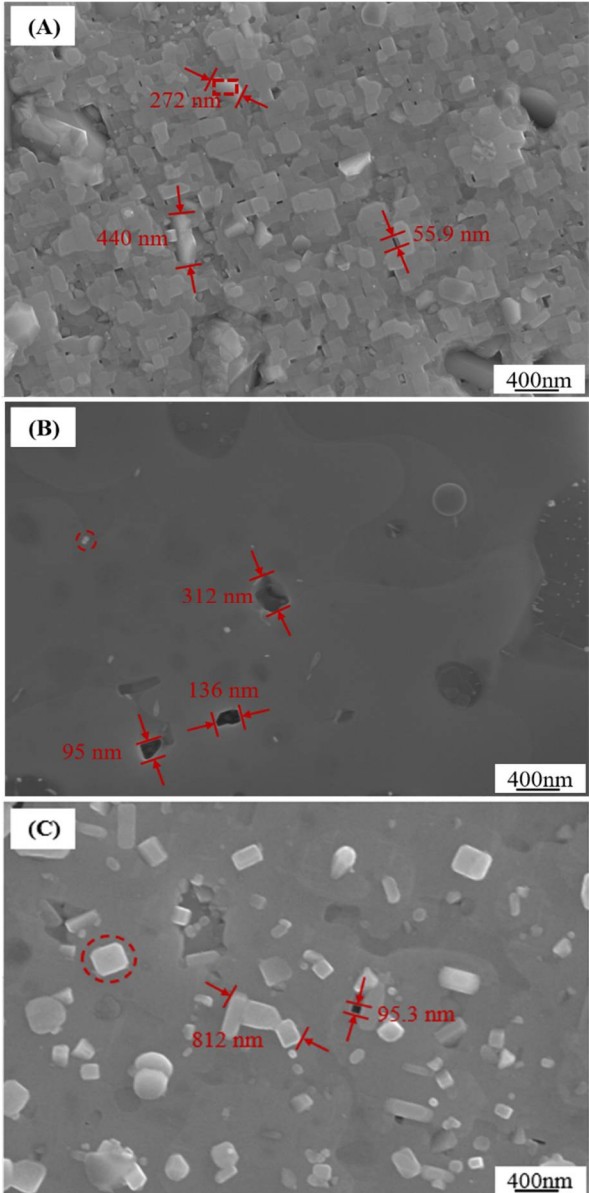

**Figure 7.** The secondary electron images of YBCO films from low vacuum pyrolysis films treated at (**A**) 810 °C and (**B**) 815 °C for 30 min, respectively. The SEM images of YBCO films from conventional film treated (**C**) at 810 °C for 1 h.

The *θ-2θ* X-ray diffraction patterns of YBCO films show (*00l*) characteristic peaks besides LaAlO$_3$ single crystal substrate peaks, which indicates that the films are *c*-axis woven YBCO superconducting films in Figure 8. As Th increased from 810 °C to 815 °C, the impurity BaCuO$_2$ phase was eliminated while (*00l*) peaks were enhanced, showing similar peak shapes to conventional films (Figure 8A). Then, we compared the peak intensity of (005) (Figure 8B). The film heat-treated at 815 °C has a higher diffraction intensity compared to 810 °C (similar thickness), suggesting a stronger c-axis grain arrangement orientation. The temperature increase may convert the randomly oriented YBCO grains to *c*-axis orientation, and the remaining Ba-Cu-O liquid phase was involved in the generation of YBCO phase [14,21].

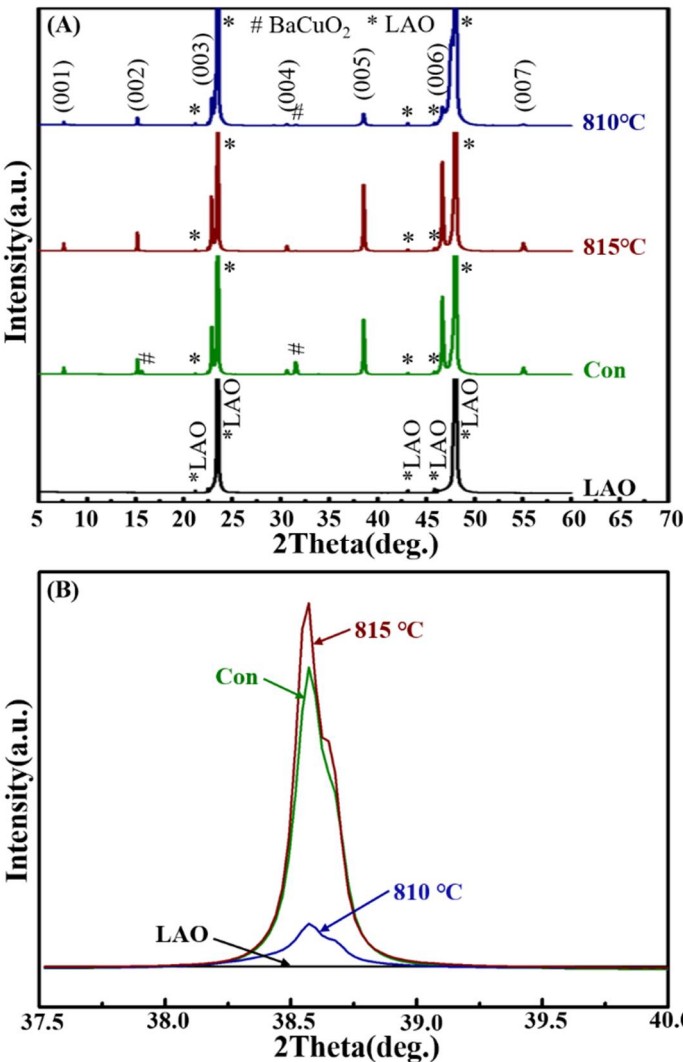

**Figure 8.** (**A**) The *θ-2θ* X-ray diffraction patterns of low-vacuum pyrolysis films after high-temperature treatment at 810 °C and 815 °C for 30 min. (**B**) Relative intensity of (005) peaks.

Raman spectra were conducted to investigate the growth orientation of YBCO films (Figure 9). The O(2,3) anisotropic vibration-related peak in the Cu-O plane of the film is at 340 cm$^{-1}$, which represents the *c*-axis out-of-plane growth of the film [23]. The peak associated with the top oxygen atom O(4) vibration is at 500 m$^{-1}$, standing for the growth in the *a*-axis [24]. As the temperature increased to 815 °C, the peak of BaCuO$_2$ weakened significantly, and O(2,3) peak was much stronger than O(4). We concluded that 815 °C was more favorable for an *a-c* biaxial weave with *c*-axis dominant growth. Also, the increase in temperature improved the homogeneity of the YBCO phase.

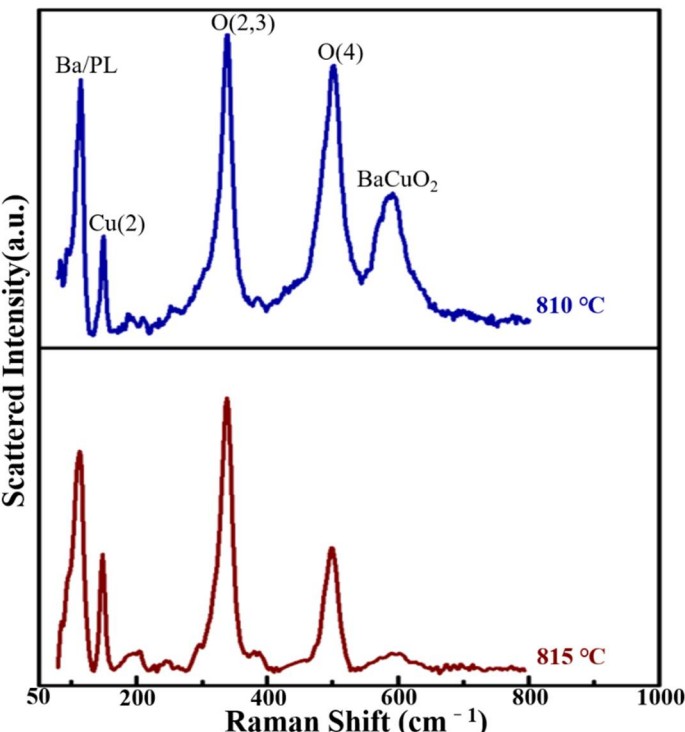

**Figure 9.** Raman spectra of low-vacuum pyrolysis films after high-temperature treatment.

The superconducting transition characteristics curves of the samples at 815 °C are shown in Figure 10. At 815 °C, $T_c$ and the transition width of YBCO superconducting film are 89.2 K and 1.4 K, respectively. The above results demonstrate a purer superconducting phase has been achieved at 815 °C (Supplementary Figure S3) [21].

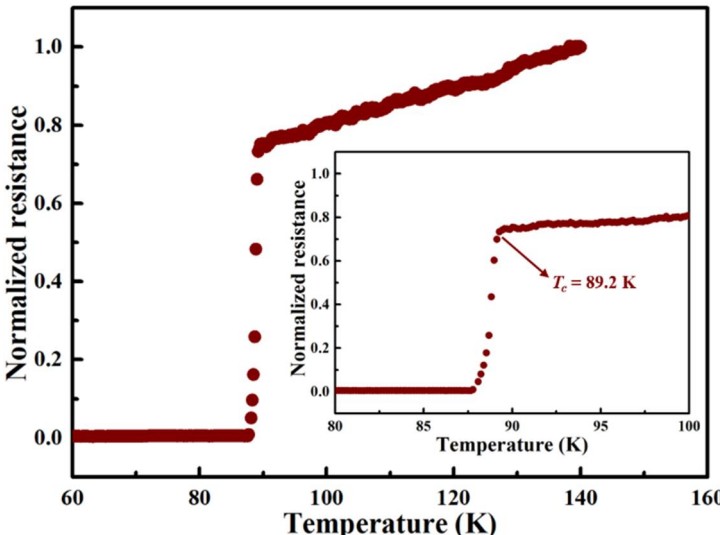

**Figure 10.** Superconducting transition characteristics (*T-R* curve) of low-vacuum pyrolysis film after 815 °C treatment. The inset is a partial enlargement of the *T-R* curve.

Finally, the $J_c$-Scan was used in the transport measurement in 77 K, self-field (Figure 11). When $T_h$ was raised from 810 °C to 815 °C, $J_c$ increased from 0.38 MA/cm$^2$ to 1.21 MA/cm$^2$, and the inset shows close $J_c$ in most areas. The above results are probably due to the significantly enhanced epitaxial growth and phase homogeneity of YBCO film.

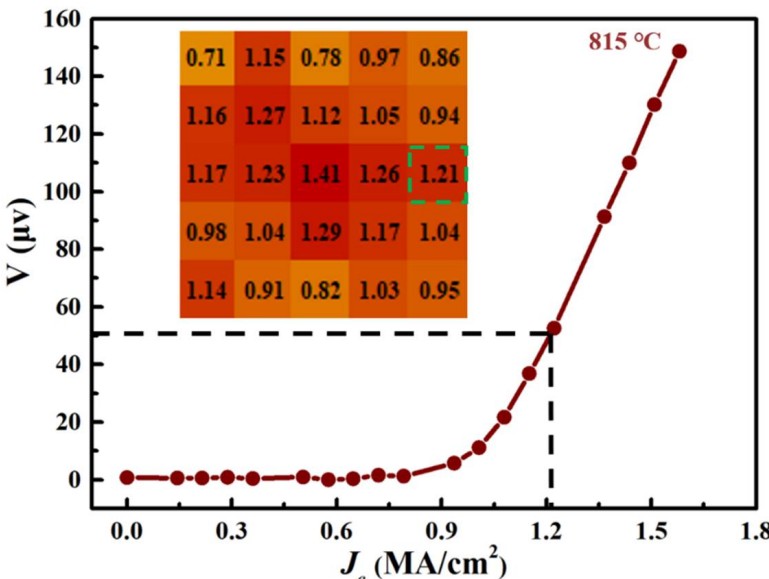

**Figure 11.** Scan-$J_c$ analysis on the middle of YBCO film after 30 min high-temperature treatment at 815 °C for low-vacuum pyrolysis film. Inset is the $J_c$ distribution diagram. The green box is $J_c$ curve.

## 4. Conclusions

This work explored a method for low-vacuum pyrolysis of YBCO superconducting film with high $J_c$. DTA/TGA analysis demonstrated that complete pyrolysis could be achieved by modulating the Y, Ba, and Cu acetate sol with a thickening agent and heating up to 600 °C. Low-vacuum pyrolysis of the wet film was accomplished by simple Ar gas depressurization, heating to 600 °C at the rates of 10–25 °C/min without introducing any carrier gas. Compared with 25 °C/min, the analysis of secondary electron images and EDS spectra demonstrate that the films were better pyrolyzed at 15 °C/min or 20 °C/min. The conventional film surface are mainly CuO particles between 150 nm and 350 nm. In contrast, the low-vacuum pyrolysis film suppresses the particle size of generally less than 50 nm, thanks to the greatly reduced pyrolysis time reduced greatly. The polycrystalline particles show close elemental ratios to the amorphous regions, which may facilitate the homogenization of the film and the generation of the YBCO phase. By increasing $T_h$ from 810 °C to 815 °C, epitaxial growth and phase homogeneity of the YBCO film were improved. As a result, $J_c$ increased to 1.21 MA/cm². The above results indicate that simple Ar gas decompression has successfully achieved efficient pyrolysis in a low vacuum environment. Low-vacuum pyrolysis film exhibits good epitaxial growth of YBCO with high critical current density. It is obvious to improve the manufacturing efficiency of YBCO coated conductors due to the simplified pyrolysis without any carrier gas applied, as well as the reduced time cost. Note that the stoichiometric ratio for the studied samples is $n$(Y):$n$(Ba):$n$(Cu) = 1.3:2:3.6. It is believed that the microstructure and the superconducting performance may be improved after optimizing the stoichiometric ratio.

**Supplementary Materials:** The following supporting information can be downloaded at: https://www.mdpi.com/article/10.3390/cryst12060812/s1, Figure S1: The preparation process of YBCO film. Figure S2: SEM images of low vacuum pyrolysis films: (A) 10 °C/min; (B) 20 °C/min. Figure S3: EDS analysis of surface particles obtained from low vacuum pyrolysis films treated at (A) 810 °C and (B) 815 °C for 30 min, respectively. (C) EDS analysis of surface particles obtained from conventional film treated at 810 °C for 1 h. Figure S4: The photo of the four-point probe method.

**Author Contributions:** Z.Y. wrote the manuscript, and performed and analyzed the DTA/TGA, XRD, Raman, Scan-$J_c$ measurements; N.C. and S.T. performed AFM, SEM and EDS. Y.L., Z.L., C.C. provided supervision. All authors have read and agreed to the published version of the manuscript.

**Funding:** This work was supported in part by the Strategic Priority Research Program of the Chinese Academy of Sciences, Grant No. XDB25000000, National Natural Science Foundation (52172271), and National Key R&D Program Project YS2021YFE030026.

**Institutional Review Board Statement:** Not applicable.

**Informed Consent Statement:** Not applicable.

**Data Availability Statement:** The data that support the findings of this study are available from the corresponding author upon reasonable request. The data that supports the findings of this study are available within the article [and its supplementary material].

**Acknowledgments:** This work was supported in part by National Key R&D Program Project YS2021YFE030026, the Strategic Priority Research Program of the Chinese Academy of Sciences, Grant No. XDB25000000, and National Natural Science Foundation (52172271).

**Conflicts of Interest:** The authors declare no conflict of interest.

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
