# Peer review of "Low-Vacuum Pyrolysis of YBCO Films by Using Fluorine-Free Metal Organic Chemical Deposition"

_crystals, doi:10.3390/cryst12060812_

Round 1

Reviewer 1 Report

The authors prepared the YBCO superconducting films using the metal-organic chemical deposition method. The authors study pyrolysis in a low vacuum environment and non-carrier gas atmosphere for the first time. They succeeded in obtaining the low vacuum pyrolysis films by simply depressurizing the air to 5 Pa, which is different from the normal pressure pyrolysis films. In this way, the authors can avoid the use of pyrolysis gases and greatly reduce the time. The quality of the film is optimized in the growth process, and the high quality of the film is demonstrated by x-ray, Raman, and SEM data.

This manuscript is clear and relevant for the field of growth of YBCO films. Their data are presented in a well-structured manner. This paper is well written. Their results are important and provide new knowledge to the field. I recommend this paper be published in Crystals.

Just a suggestion. It would be perfect if the authors could show the resistivity and susceptibility data to further demonstrate the quality of the films.

Minor:

Line50: pa—>Pa

Reviewer 2 Report

Short time pyrolysis under low gas pressure in the synthesis of YBCO films by the fluorine free MOD method are reported in this paper.  There are several points to be revised before evaluating quality of the paper.

Addition of explanation on the atmosphere of TG/DTA analysis is needed to understand decomposition behavior of the organic acids.

How to keep low-vacuum condition at 5 Pa during decomposition, which accompanies generation of CO2 and H2O gases, up to 600°C ?  This is the most important point when one traces this method.

BaCuO must be BaCuO2.

In “Materials Methods”, does Four-lead method mean four-point probe method?

For Figure 10,

Which method is used in the transport measurement, four-point probe method or Jc-Scan?

No description of measured temperature.

Vertical axis should be V (µV) or E (µV/cm).

If possible, Jc values are compared with those in the previous reports. 

In line 214, “intergranular connectivity” means polycrystalline films, while the film sintered at 815°C seems epitaxial grown single crystalline one.

"Secondary electron images" is more suitable than "SEM images". 

“Tokyo University” should be “University of Tokyo” or “Motoki et al.”, because more than 20 “Tokyo Universities” exist in Japan.

Round 2

Reviewer 1 Report

The authors have improved the quality of this paper following my suggestions. I recommend this paper be published in Crystals.

Author Response

We earnestly appreciate the reviewer's warm work.

Reviewer 2 Report

The revised paper satisfies the referee's requests and became a suitable one for publication. Before publication, following minor points should be improved.

line 69: Superconducting ->superconducting

line 74: Argon -> argon

lines 121, 174, 188, 235: Secondary -> secondary

line 217: 89.2 k and 1.4 k -> 89.2 K and 1.4 K
